# MULTILIFEQA: A MULTIDIMENSIONAL LIFESTYLE QUESTION ANSWERING BENCHMARK FOR COMPREHENSIVE HEALTH REASONING WITH LLMS

## ABSTRACT

Recent advances in wearable devices and mobile sensing technologies have enabled the continuous collection of multimodal lifestyle data. However, transforming these heterogeneous signals into coherent and interpretable insights for health management remains a fundamental challenge. These difficulties arise both at the data level, where signals are fragmented and lack a unified structure, and at the modeling level, where existing methods are often limited to single domains and short-term tasks. Large language models (LLMs) have demonstrated strong potential for complex reasoning, yet systematic benchmarks to evaluate their cross-dimensional and long-horizon reasoning abilities in lifestyle health are still lacking. We propose MultiLifeQA, the first large-scale QA dataset and benchmark for multidimensional lifestyle health reasoning. MultiLifeQA spans four lifestyle dimensions (diet, activity, sleep, and emotion) and contains 22,573 questions across single-user and multi-user scenarios. The tasks are grouped into five categories, spanning from simple fact retrieval to complex cross-dimensional temporal reasoning, providing a comprehensive evaluation of model reasoning capabilities. We establish two prompt evaluation methods: context and database-augmented, along with fine-grained metrics that evaluate query validity, execution quality, and final answer accuracy. Extensive experiments on eight open-source and three proprietary LLMs highlight both the capabilities and limitations of current models in long-term, multidimensional health reasoning. By addressing this gap, Multi-LifeQA establishes a standardized benchmark that advances LLMs toward more integrated health analytics and personalized interventions. The code and datasets are publicly available at https://anonymous.4open.science/r/MultilifeQA-05D2.

## 1 INTRODUCTION

Analyzing lifestyle behaviors and delivering timely, personalized feedback are essential for promoting effective health management and preventing disease. Noncommunicable diseases (NCDs) such as heart disease, cancer, and diabetes account for 75% of global deaths, causing over 43 million deaths annually (World Health Organization, 2023). Insufficient physical activity, poor sleep, unhealthy diets, and chronic psychological stress are established risk factors for NCDs onset and progression (World Health Organization, 2023; St-Onge et al., 2016; Vaccarino et al., 2025). Timely and accurate identification of lifestyle factors, coupled with their translation into personalized, actionable recommendations, can substantially reduce disease risk and prevent premature mortality (Chu et al., 2016; Motevalli & Stanford, 2025). Wearable devices and smart applications have made the continuous, fine-grained collection of daily-life data increasingly convenient (Jamieson et al., 2025). Many smartwatches (e.g., Apple (Apple Inc., 2025) and Google (Google LLC, 2025)) capture step counts, calorie expenditure, heart-rate variability, and sleep stages. Applications can quantify diet from images (Oei et al., 2024), classify activity from IMU signals (Zareeia et al., 2025), and estimate stress from electrodermal activity (EDA) and heart-rate recovery (McDuff et al., 2025). These advancements provide a rich multimodal data foundation for health management.

Yet, despite this wealth of information, transforming heterogeneous multimodal signals into interpretable health insights and delivering feedback through natural language question answering (QA) remains a major challenge. This challenge stems primarily from two factors: **First, the limitations**

**at the data level.** Raw lifestyle data are often stored in fragmented forms such as log files and high-frequency time-series signals. These data lack a unified structure and are not directly interpretable. For ordinary users, it is nearly impossible to extract meaningful health insights from large collections of step counts, heart rate fluctuations, or sleep-stage logs. Instead, users expect to obtain an analysis of their health conditions in an intuitive way, such as by asking some questions: "Has my deep sleep duration been continuously decreasing over the past week?" or "Did my stress decrease when I increased the time I spent on aerobic exercise?" **Second, the limitations of current model level.** Effective health reasoning often requires integrating multiple dimensions and temporal dynamics to detect abnormal patterns and capture long-term trends. For instance, a single night of reduced sleep may seem trivial, but when paired with rising stress and declining activity over weeks, it reveals a potential health risk that single-dimensional analysis would miss. Traditional machine learning and deep learning methods typically focus on single-dimensional prediction or classification tasks, such as activity recognition from accelerometer signals or predicting nightly sleep quality based on heart rate variability. While these approaches perform well on individual tasks, they lack the capacity for multi-dimensional reasoning across heterogeneous lifestyle factors. As a result, they are limited in supporting continuous, holistic health assessment and personalized management.

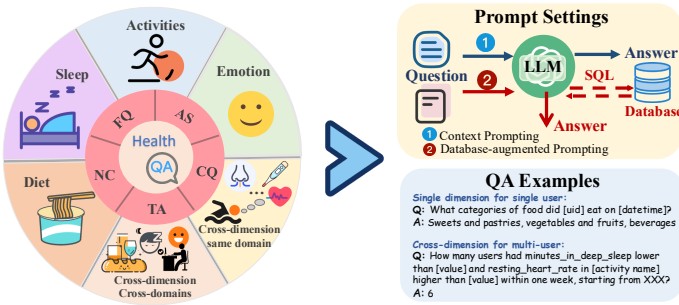

Figure 1: MultiLifeQA: A large-scale health reasoning QA benchmark, ranging from single-dimensional to cross-dimensional, single-user to multi-user tasks. It covers four core domains: diet, emotion, sleep, and activity, and has five categories questions: Fact Query (FQ), Aggregated Statistics (AS), Numeric comparison (NC), Conditional Query (CQ) and Trend Analysis (TA).

To systematically evaluate and advance models capable of multidimensional health reasoning, robust QA benchmarks are essential. In recent years, several health-related QA datasets have been introduced, including those focused on nutritional decision making (NGQA (Zhang et al., 2024)), activity and sensor data analysis (SensorQA (Reichman et al., 2025)), sleep health (SleepQA (Bojic et al., 2022)), emotional and psychological support (MentalChat16K (Xu et al., 2025) , MHQA (Racha et al., 2025)), and lifelog analysis (OpenLifelogQA (Tran et al., 2025)). These efforts offer valuable insights into applying the QA paradigm to health and lifestyle analysis. However, they are typically limited to single domains or narrowly defined tasks and do not capture the interactions across multiple lifestyle dimensions. In addition, the emergence of large language models (LLMs) offers a promising avenue to overcome model-level limitations. Recent studies have explored the use of LLMs for personalized health and lifestyle analysis, including sleep assessment (Khasentino et al., 2025; Fang et al., 2024), activity prediction (Kim et al., 2024; Yu et al., 2025), daily logs generation (Tian et al., 2025), and emotion analysis (Xu et al., 2024; Yang et al., 2024). Although these studies provide initial evidence of LLMs' applicability in health and lifestyle analysis, they generally evaluate performance within single dimensions or narrowly defined tasks. To date, there is no unified benchmark for systematically assessing LLMs' ability to perform long-horizon reasoning and integrated analysis across multiple dimensions, including diet, activity, sleep, and emotion.

To address this gap, we present MultiLifeQA, as illustrated in Figure 1, the first large-scale QA benchmark constructed from multidimensional lifestyle data. It contains 22,573 questions, covering tasks that range from basic fact retrieval to cross-dimensional, long-horizon reasoning. We develop a systematic evaluation framework with two settings: *Context Prompting (CP)*, in which the model answers directly from prompts containing the relevant data and questions, and *Database-augmented Prompting (DP)*, in which the model generates and executes SQL queries and subsequently performs reasoning based on the returned results. We also propose a set of metrics to evaluate the fine-grained

performance of the model reasoning process: *Accuracy* for final-answer correctness, *SQL Validity (VA)* to assess whether the generated query is complete and executable, *Execution Accuracy (EX)* to measure whether the execution result contains all correct information, and *Acc/EX*, the ratio of cases where the LLM infers the correct answer given that the SQL query executes successfully.

We conduct a systematic evaluation of eight open-source and three proprietary models on MultiLifeQA, and provide an in-depth analysis. The results indicate that first of all, proprietary models generally outperform open-source ones: GPT-4o achieves the highest accuracy 57.02% with CP, while Gemini leads with DP (39.04%). Second, in terms of question types, aggregation statistics are the most challenging, with average accuracies of only 5.98% (CP) and 14.2% (DP), highlighting the limitations of current LLMs in long-horizon reasoning. Third, the comparison across answer types shows that LLMs perform well on questions with Boolean or single-number answers, but their accuracy drops substantially for more complex reasoning tasks that involve multiple-item answers. Last but not least, the experiment show that cross-dimensional and multi-user reasoning tasks are significantly more challenging. With CP, average accuracy drops from 41.54% (single-dimension) to 23.42% (cross-dimension). With DP, the decline is even sharper, from 30.84% to 9.63%. Meanwhile, the performance of all models on multi-user tasks generally lags behind that on single-user tasks (16.99% vs. 24.43%). Overall, these results suggest that while current LLMs possess capability for health reasoning, substantial limitations remain in cross-dimensional and multi-user aggregate reasoning, underscoring important directions for future research.

| Method | Task | Scale | Annotation | Covered Dimensions | Multi-user | Cross-dimension |
|---|---|---|---|---|---|---|
| NGQA (Zhang et al., 2024) | Nutrition reasoning | 13.8K | LLM & human validation | Nutrition Health | ✗ | ✗ |
| SensorQA (Reichman et al., 2025) | Daily-life QA | 5.6K | Manual Creation | Activity and Location | ✗ | ✗ |
| SleepQA (Bojic et al., 2022) | Sleep guidance | 7K | Manual Creation | Sleep data | ✗ | ✗ |
| MentalChat16K (Xu et al., 2025) | Mental health dialogue | 16.1K | Interview Collection & Synthetic | Emotion / Mental health | ✗ | ✗ |
| OpenLifelogQA (Tran et al., 2025) | Lifelog QA | 14.2K | LLM Ggeneration & Manual Creation | Multi-modal lifestyle | ✗ | ✓ |
| **MultiLifeQA (Ours)** | **Cross-dimensional health QA** | **22.6K** | **Template Generation & Human Validation** | Diet, activity, sleep and emotion data | ✓ | ✓ |

Table 1: Comparison of existing QA dataset benchmarks and MultiLifeQA.

## 2 RELATED WORK

**Lifestyle Datasets for Health Analysis.** Lifestyle datasets are essential for monitoring health behaviors and supporting disease prevention. Existing resources capture diverse lifestyle aspects but often emphasize short-term or single-dimension monitoring. For instance, MMASH (Rossi et al., 2020) provides 24-hour multimodal data from 22 participants for sleep and psychological analysis, WESAD (Philip Schmidt et al., 2018) records stress and affective states in controlled settings, and CAPTURE-24 (Doherty et al., 2017) offers large-scale accelerometer data with sleep diaries for activity recognition. More recent efforts extend to longer-term and multidimensional monitoring, such as LifeSnaps (Yfantidou et al., 2022), GLOBEM (Xu et al., 2022), ETRI Lifelog (Oh et al., 2025). Among them, AI4FoodDB (Romero-Tapiador et al., 2023; Lacruz-Pleguezuelos et al., 2025) stands out for its comprehensive design, collecting one month of multimodal lifestyle and clinical data from 100 participants. Covering nutrition, activity, sleep, emotion, and other health dimensions, it uniquely supports cross-dimensional analysis and long-term health trajectory modeling. Therefore, we adopt AI4FoodDB as the source dataset for constructing our QA benchmark.

**Health Lifestyle QA Benchmarks.** Recent studies have applied LLMs to personalized health and lifestyle tasks such as sleep and fitness guidance (Khasentino et al., 2025), dietary assessment (Hua et al., 2024), daily activity query (Yu et al., 2025), mental health analysis (Xu et al., 2024; Yang et al., 2024), and lifelogs generation (Tian et al., 2025), demonstrating their potential for interpreting personal health data. To further enhance LLMs' capabilities in health analysis and reasoning, several QA datasets tailored to personal health and lifestyle analysis have been developed. For instance, NGQA (Zhang et al., 2024) models dietary decision-making with graph-based reasoning for personalized nutrition, and SensorQA (Reichman et al., 2025) interprets raw sensor data through QA. Other resources include SleepQA (Bojic et al., 2022) for sleep guidance, MentalChat16K (Xu et al., 2025) for conversational emotional well-being support, and OpenLifelogQA (Tran et al., 2025) for lifestyle queries derived from personal lifelogs. As summarized in Table 1, existing datasets, despite these advances, remain limited to single dimension (e.g, sleep) and single-user, and also provide little support for long-horizon reasoning. To address this gap, we present the first large-scale QA benchmark built on a comprehensive multidimensional lifestyle dataset, enabling systematic evaluation and advancement of LLMs in long-term, multi-user, and cross-dimensional health reasoning.

## 3 MULTILIFEQA

### 3.1 DATASET OVERVIEW

MultiLifeQA consists of 22,573 questions spanning four lifestyle domains: diet, sleep, activity, and emotion, including 13,452 single-user queries, which focus on reasoning about the lifestyle of a single individual, and 9,121 multi-user queries, which involve comparisons or joint reasoning across multiple individuals. Effective health analysis requires both low-level descriptive retrieval and high-level complex reasoning. Driven by this motivation, MultiLifeQA organizes reasoning tasks into five major categories: Fact Query, Aggregated Statistics, Numeric comparison, Conditional Query, and Trend Analysis, with their distribution illustrated in Figure 2c. Specifically, *Fact Query* establishes baseline information to reconstruct individual behavioral trajectories; *Aggregated Statistics* extends analysis to longer temporal windows for characterizing long-term behavioral patterns; *Numeric comparison* reveals relative differences and individual preferences; *Conditional Query* incorporates personalized thresholds and group-level references to identify anomalies and potential risks; and *Trend Analysis* captures dynamic changes, helping to uncover emerging health concerns or signs of continuous improvement. Overall, these categories reflect the complex, multi-dimensional aspects of lifestyle reasoning and establish a direct link between raw behavioral data and health-related insights. The answer types mainly include categorical responses (Yes/No), numerical values (single-number), short text (one word or phrase), pairwise answer (=2 items), and multi-item answer (≥3 items). In addition, we visualize the distribution of meaningful lexical items extracted from the questions, as illustrated in Figure 2b. Terms such as sleep, stress, active, and oxygen saturation appear most frequently, highlighting people's primary concerns about lifestyle and health.

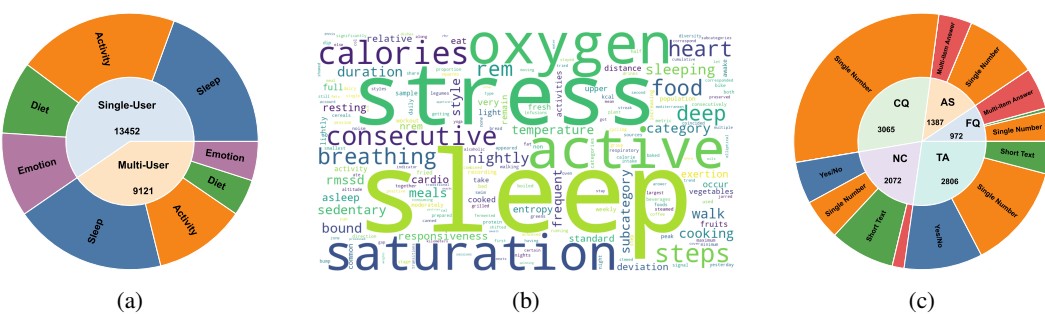

(a)                                    (b)                                    (c)

Figure 2: (a).User and single domain distribution. (b).Visualization of word frequency in the health question. (c).Question and answer distribution.

### 3.2 DATA SOURCE

The source data of MultiLifeQAare derived from AI4FoodDB (Romero-Tapiador et al., 2023), a large-scale personal lifestyle database collected from 100 participants over a one-month period. It integrates self-reported questionnaires (e.g., surveys on lifestyle habits), clinical assessments (e.g., standardized physical examinations and laboratory test results), and continuous digital records from wearable devices (e.g., step counts, heart rate, sleep patterns, and activity levels). Collectively, these data span a wide range of domains, including anthropometrics, lifestyle and health history, nutrition, biomarkers, gut microbiome, vital signs, physical activity, sleep behaviors, and emotional states. Owing to its multidimensional and comprehensive design, AI4FoodDB represents one of the most complete open resources currently available for studying lifestyle factors and their interactions with health outcomes. Building on this foundation, we design a pipeline that automatically generates health queries, from simple fact retrieval to long-horizon cross-dimensional reasoning.

### 3.3 QA DATASET GENERATION PIPELINE

We design an automated pipeline to generate MultiLifeQA, as illustrated in Figure 3. In particular, we develop a scalable code framework that can be flexibly extended with new data and reasoning tasks, with detailed instructions and guidelines provided in Appendix D. The main steps for generating MultiLifeQA are outlined as follows.

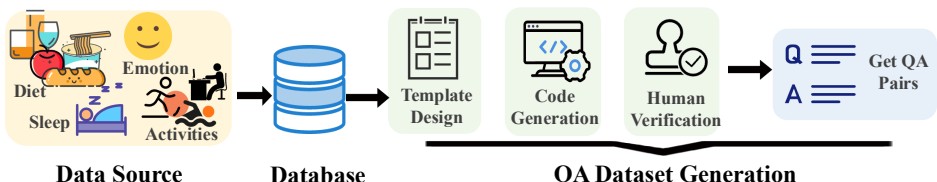

Figure 3: MultiLifeQA Dataset Generation Pipeline, which transforms raw lifestyle data from four domains (diet, sleep, activity, and emotion) into high-quality QA pairs through database construction, template design, automated code generation, and human verification.

**Database Construction.** We build a structured database on MySQL, importing data on diet, physical activity, sleep, and emotion from the source data. Records are aligned by anonymized user identifiers and timestamps to support subsequent question generation and ground-truth validation.

**Template Design.** We design question templates at both single-user and multi-user levels to cover individual reasoning and multi-user-comparison health reasoning tasks. Concretely, we first hand-craft single-dimension lifestyle templates for five task categories (Fact Query, Aggregated Statistics, Numeric comparison, Conditional Query, Trend Analysis), targeting reasoning needs related to one dimension (diet, activity, sleep, or emotion). We then extend these templates to generate cross-dimensional composite queries that capture complex interactions across lifestyle dimensions and reveal latent health patterns and deep insights. The templates are designed to ensure both scalability and interpretability, with further details provided in Section 3.4.

**Code Generation.** Given the database schema and designed templates, we develop a programmatic framework that automatically instantiates natural-language questions from templates and retrieves corresponding answers via SQL queries and subsequent computations as ground truth, ensuring diversity without duplication and balanced coverage across question types.

**Human Verification.** We manually inspect all of the generated questions and ground-truth answers, removing invalid or duplicate items, thereby yielding a high-quality QA dataset.

## 3.4 QUERY TEMPLATE DESIGN

**Single-dimension Template Design.** The single-dimension templates are designed to capture lifestyle patterns and enable health reasoning across four domains: diet, physical activity, sleep, and emotion. Together, these domains represent fundamental determinants of health: diet reflects nutritional balance and eating habits; activity captures daily energy expenditure and exercise behaviors; sleep signals indicate restorative quality and physiological stability; and emotion data reveal stress states and their drivers. Collectively, they provide the foundation for understanding individual behaviors and health trajectories. We design QA templates according to five categories, with representative examples provided in the Appendix G. These templates convert raw lifestyle logs into interpretable questions, enable reasoning across multiple time scales, and facilitate the identification of potential health risks or trends for improvement. Yet, single-dimension queries cannot fully capture the complex associations among health determinants, so we extend our approach to cross-dimension, enabling queries that capture interactions among diet, activity, sleep, and emotion, and thus support higher-level reasoning tasks more aligned with real-world health management.

**Cross-dimension Template Design.** We extend the templates to generate cross-dimensional queries that explicitly model interactions among lifestyle factors. For example, physical activity levels may influence sleep quality; diet may affect emotion; and emotion, in turn, can modulate dietary and sleep behaviors. Following these observations, many user-centric questions naturally arise, such as: "When I increase aerobic exercise, do I experience longer deep sleep and lower stress?" and "Are specific food categories or cooking methods associated with sleep quality?" We formalize these questions as computable queries and organize them into five categories (see detailed examples in Appendix G.1). *Fact Query* captures cross-dimensional snapshots of multiple lifestyle factors on a given date. *Aggregated Statistics* reveals long-term associations, for example, correlations between sleep quality and exercise regimes. *Numeric comparison* assesses relative differences across dimensions, such as whether weeks with higher activity levels show lower stress. *Conditional Query*

detects joint threshold events, such as days where sedentary time is high and stress is elevated. Finally, *Trend Analysis* tracks dynamic co-variation over multiple days, identifying patterns such as co-occurrences of reduced activity, insufficient sleep, and rising stress. Therefore, these queries enable systematic evaluation of cross-dimensional, long-horizon health reasoning.

**Multi-user Template Design.** While single-user queries can capture an individual's lifestyle characteristics and health trajectory, their scope remains confined to the personal level and does not provide comparison and reference among more users. Many health reasoning tasks become more meaningful when contextualized within a broader population, such as evaluating an individual's sleep quality relative to age-matched peers or community-level averages. Similarly, ranking activity levels within a peer group can inform personalized adjustments. To support such analyses, we design multi-user queries that allow comparison, aggregation, and filtering across individuals, thereby uncovering health insights that are not only more generalizable but also socially contextualized.

For implementation, we utilize anonymized user identifiers and align behavioral records across individuals through shared timestamps, enabling synchronized and systematic cross-user comparisons. Multi-user queries adopt the same five reasoning categories as single-user queries but place greater emphasis on group-level statistics and contrasts. For example: *Fact Query* can be used to retrieve the user with the longest activity duration on a given day; *Aggregated Statistics* computes the average REM sleep duration across all users in a week; *Numeric comparison* quantifies the difference between a specific user and the multi-user mean; *Conditional Query* filters subgroups of users with insufficient sleep over consecutive days, and *Trend Analysis* captures multi-user-level dynamics over time. Some representative examples are provided in the Appendix G.2. Thus, MultiLifeQA can support cross-dimensional multi-user queries, such as examining whether higher energy expenditure aligns with longer deep sleep or whether frequent fried-food consumption correlates with elevated stress, revealing broader health insights that link individual behaviors with group-level trends.

**Summary.** MultiLifeQA is a comprehensive health reasoning dataset that encompasses both single-user and multi-user scenarios, supporting cross-dimensional and long-horizon reasoning. It enables the evaluation of models' ability to capture fine-grained individual lifestyle characteristics, while simultaneously assessing their reasoning capabilities across multiple dimensions and at the multi-user level. By combining these features, MultiLifeQA establishes a systematic and robust benchmark that can drive advances in health analytics and support the development of personalized interventions.

## 4 EXPERIMENTS

We define two evaluation settings and perform a comprehensive evaluation of eight widely used open-source LLMs and three proprietary LLMs on MultiLifeQA. We first compute and compare the overall accuracy of all models across the complete set of reasoning questions. Subsequently, we perform a detailed analysis along multiple dimensions, including comparisons by question and answer distribution, as well as differences across dimensions and user settings.

### 4.1 EXPERIMENTAL SETUP

**Evaluation Settings.** To assess the capability of mainstream LLMs for comprehensive health analysis and reasoning on MultiLifeQA, we establish two evaluation settings: *Context Prompting*, which directly embeds user-specific data into the prompt after pre-filtering, and *Database-augmented Prompting*, which leverages structured SQL queries to retrieve relevant information before reasoning. The reason we adopt these two complementary settings is that context prompting is the most straightforward and lightweight strategy in existing work (Lee et al., 2024), embedding data directly into the prompt with 'zero engineering cost,' and is widely used as a fair baseline for comparison. However, when tasks involve larger-scale reasoning data, directly embedding all information into the prompt often exceeds the context window of LLMs. Database-augmented prompting provides a feasible solution by leveraging structured SQL queries to effectively support complex reasoning (Zhu et al., 2024). Therefore, we employ both settings to comprehensively evaluate LLMs' capabilities in health reasoning tasks on MultiLifeQA.

*1) Context Prompting.* In this setting, the question and the relevant health data for the target user are embedded directly into the prompt for the LLM to answer. To mitigate the context-length limitations of LLMs, we pre-filter the data during prompt construction by first selecting a user identifier and

retaining only the data corresponding to this user for inclusion in the prompt. Note that even with pre-filtering, a small number of questions still exceed the context window due to the large volume of data required for reasoning. Therefore, for multi-user reasoning tasks with much larger data scopes, we design database-augmented prompting.

*2) Database-augmented Prompting.* In this setting, we first encode the question and the constructed database schema into a carefully designed prompt template to guide the LLM in generating the corresponding SQL query. The generated SQL is then subject to a preliminary check to ensure it is a complete `SELECT` statement; otherwise, it is considered invalid to the database and directly marked as a failure. The system then executes the passed SQL, and if execution raises an error it is also treated as a failure; otherwise, the results are returned to the LLM, which performs further reasoning on the feedback to produce the final output. This approach can be evaluated over the entire dataset, supporting both single-user and multi-user reasoning tasks.

**Prompt Design.** We design tailored templates for the two settings. For *1) Context Prompting*, the prompt consists of four components: an overall task description, the reasoning question, the relevant data, and the specification of the expected answer type. For *2) Database-augmented Prompting*, the prompt is structured in two stages. In the *(i) SQL Generation Stage*, it includes the overall task description, the reasoning question, the schema of the relevant database tables, and any explicit database constraints. In the *(ii) Answer Generation Stage*, it consists of the overall task description, the reasoning question, the generated SQL, the results returned from executing the SQL query, and the specification of the expected answer type. More details are provided in Appendix E.

**LLMs.** We evaluate current mainstream LLMs, encompassing both open-source and proprietary models, to provide a comprehensive and representative assessment.

*1) Open-source models.* We include Llama-3.2-3B-Instruct, Llama-3.1-8B-Instruct, and Llama-3.1-70B-Instruct (4-bit) (Grattafiori et al., 2024); Phi-3.5-mini-instruct (Abdin et al., 2024); Mistral-7B-Instruct-v0.3 (Jiang et al., 2023); DeepSeek-R1-Distill-Qwen-7B (Guo et al., 2025); and the Qwen-2.5 series, including Qwen-2.5-7B-Instruct, Qwen-2.5-14B-Instruct (4-bit and 8-bit), and Qwen-2.5-32B-Instruct (4-bit) (Hui et al., 2024).

*2) Proprietary models.* We further include leading proprietary LLMs: GPT-4o (Achiam et al., 2023), Claude-3-haiku (Anthropic AI, 2024), and Gemini 2.5 Lite (Flash-Lite) (Comanici et al., 2025).

**Evaluation Metrics.** We use accuracy as the primary metric to evaluate LLMs performance on reasoning tasks. For the *database-augmented prompting*, we further introduce three additional, finer-grained metrics to provide a more comprehensive assessment of LLMs performance: *1) Accuracy*: For each answer type, we adopt tailored evaluation criteria to ensure fairness and precision. For yes/no answer type, the prediction must match the ground truth. For numeric answer type, a tolerance is allowed: when the ground-truth answer is an integer ($\leq 14$), an absolute error of at most $\pm 1$ is permitted; when the answer is a integer ($> 14$) or a real number, an error bound of $\max(0.5\% \cdot |gt|, 0.01)$ is permitted. This criterion preserves tolerance for small integers while maintaining precision for large integers and real-valued answers. For multi-item answers, the prediction and the ground truth must contain the same number of items, and each corresponding item must be correct. *2) SQL Validity (VA)*: The proportion of generated SQL queries that pass a preliminary check (ensuring a complete `SELECT` statement) and execute on the database without errors. *3) Execution Accuracy (EX)*: We define *EX* as the proportion of generated SQL queries that can be successfully executed and whose results provide all the information required to derive the correct answer (within the tolerance defined in Accuracy). *4) Acc/Execution Accuracy (Acc/EX)*: The accuracy of the LLM's final answers conditioned on EX, i.e., the proportion of correct answers given that the SQL query executed successfully and returned the appropriate intermediate results.

## 4.2 RESULTS AND DISCUSSION

**Overall Results.** Table 2 and Figure 4 summarize the performance of all LLMs on MultiLifeQA. First, the results indicate that proprietary models (GPT-4o, Gemini 2.5 Lite, Claude-3-Haiku) generally outperform open-source models under both evaluation settings. With *Context Prompting*, GPT-4o achieves the highest accuracy of 57.02%. With *Database-augmented Prompting*, Gemini attains the best accuracy at 39.04%. Smaller models (e.g., deepseek-coder-1.3B) fail to complete the tasks, while medium-to-large open-source models (e.g., Qwen-2.5-7B, Llama-3.1-70B) perform

reasonably well when using *Context Prompting* but accuracy drops substantially with *Database-augmented Prompting*, indicating obvious gaps of current LLMs in complex reasoning.

Secondly, with the database-augmented setting, the main limitation arises from the models' ability to generate SQL queries that are both executable and semantically accurate. As shown in Table 2, all models exhibit low execution accuracy (EX), with an average of only 25.94%. On the other hand, once the SQL executes correctly and returns the requisite information (satisfied EX), models are relatively reliable at inferring the final answer: seven models achieve Acc/EX above 70%, and GPT-4o reaches 95.65%. These findings indicate that for cross-dimensional, multi-user, and long-horizon health reasoning, the integration of external tools such as relational databases is both effective and essential. However, accurately interpreting database schemas, understanding inter-table relationships, and generating executable and precise SQL queries remain key challenges for current LLMs.

Moreover, we investigate the effects of model size and quantization on performance within the same model series, using Qwen-2.5 as an example. Detailed results are provided in Appendix H.2. Overall, increasing model size leads to substantial performance gains. As Qwen-2.5 scales from 7B to 32B parameters, overall accuracy improves consistently, rising from 40.45% to 53.86% when using *Context Prompting* and from 21.45% to 26.95% with *Database-augmented Prompting*. These results indicate that larger models possess stronger capabilities for health reasoning. Moreover, quantization precision also affects performance. For Qwen-2.5-14B, 8-bit quantization achieves higher accuracy than 4-bit, indicating that higher-precision quantization better preserves reasoning capabilities while maintaining efficiency and storage benefits.

Table 2: Overall results of all LLMs on MultiLifeQA.

| Dataset | Context Prompting | Database-augmented Prompting | | | |
|---|---|---|---|---|---|
| Metrics | Acc (%) | Acc (%) | VA(%) | EX(%) | Acc/EX(%) |
| **Open Source LLMs** | | | | | |
| deepseek-coder-1.3B | 1.09 | 1.26 | 43.83 | 14.62 | 4.00 |
| Llama-3.2-3B | 20.18 | 13.47 | 47.29 | 17.99 | 67.68 |
| Phi-3.5-mini-3.8B | 20.57 | 16.16 | 56.67 | 20.23 | 77.34 |
| Mistral-v0.3-7B | 30.97 | 9.03 | 28.46 | 11.48 | 75.35 |
| Qwen-2.5-7B | 40.45 | 21.45 | 55.83 | 24.71 | 84.78 |
| Llama-3.1-8B | 20.65 | 21.53 | 63.33 | 27.25 | 77.73 |
| gemma-2-IT-9B | 24.44 | 14.54 | 56.24 | 27.85 | 51.15 |
| Llama-3.1-70B | 40.51 | 13.91 | 45.77 | 22.73 | 58.41 |
| **Proprietary LLMs** | | | | | |
| Gemini 2.5 Lite | 44.81 | **39.04** | **84.84** | **45.97** | 82.92 |
| Claude-3-haiku | 35.30 | 29.30 | 74.06 | 36.49 | 75.71 |
| GPT-4o | **57.02** | 34.71 | 63.85 | 35.97 | **95.65** |

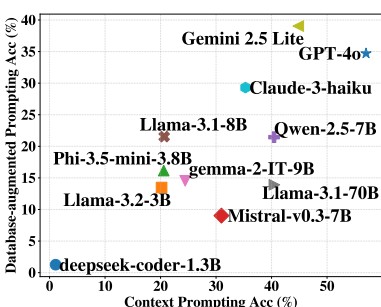

Figure 4: Accuracy (%) of all LLMs under two evaluation settings.

**Results by Question Type and Answer Type.** To further analyze the differences in model performance on questions of different reasoning difficulties, we analyze the reasoning results of different question types and answer types. Appendix H.1 reports the detailed results. Overall, from the perspective of question types, LLMs perform best on trend analysis and conditional query tasks. With *Context Prompting*, the average accuracies reach 56.08% and 48.03%, respectively. In contrast, aggregation statistics questions are the most challenging category: accuracy is only 5.98% with *Context Prompting*, and although it rises to 14.29% with *Database-augmented Prompting*, it remains the lowest, highlighting clear limitations in long-horizon reasoning for current LLMs. From the perspective of answer types, models perform best on questions whose answers are Yes/No (49.69%) and Single Number (45.25%) with *Context Prompting*, indicating that simple answer formats that map directly to a Boolean or a single numeric value are easier to handle. In contrast, for more complex reasoning tasks with pairwise answers and multi-item answers, accuracy drops substantially. With *Context Prompting*, the averages are 12.44% and 8.12%, and with *Database-augmented Prompting*, they are only 4.29% and 3.21%, respectively.

To complement these overall findings, we further examine the performance of the best model, GPT-4o, as shown in Figure 5, across question and answer types. The results show that GPT-4o's reasoning accuracy is substantially higher than the average across all models, with the highest results on Conditional Query and Fact Query: with *context prompting*, the accuracies reach 71.5% and 69.8%, respectively. In the *database-augmented setting*, when the generated SQL executes successfully, Acc/EX rises to 99.3% (CQ) and 93.7% (FQ). By answer type, GPT-4o performs best on short-text answers. In particular, with *Database-augmented Prompting*, when SQL returns the correct answer, Acc/EX reaches 99.2%. However, even if GPT-4o performs outstandingly on these tasks, GPT-4o

remains weak on pairwise-answer and multi-item answer questions, with accuracy only of 33.9% and 28.1%, respectively. A deeper analysis shows that in these challenging tasks, Acc drops much more than Acc/EX, underscoring that understanding data relations and generating executable SQL remain key bottlenecks for current LLMs when faced with complex reasoning.

In summary, these findings indicate that contemporary LLMs perform relatively well on fact retrieval and comparison questions as well as for simple answer formats (e.g., yes/no, single number), but they still struggle with aggregate statistics and more complex answer forms, highlighting the necessity for future research to improve in long-horizon reasoning and complex answer generation.

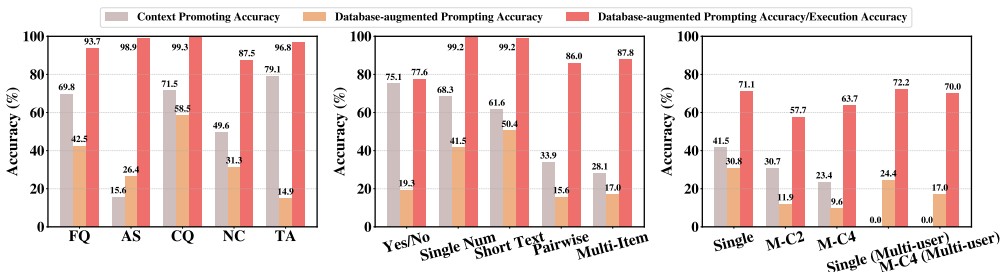

Figure 5: Accuracy of GPT-4o by question type (**left**) and answer type (**middle**). Average accuracy of all models across different dimensions and user settings (**right**).

**Results with Varying User Settings and Dimensions.** We evaluate and analyze model performance on single- and cross-dimensional reasoning tasks, as well as the impact of user settings. Detailed results are provided in Appendix H.3. Figure 5 reports the average performance across dimensions for all models. "Single" denotes single-dimension tasks confined to one lifestyle domain (diet, activity, sleep, or mood). "M-C2" denotes cross-dimensional reasoning over two distinct domains (e.g., jointly analyzing sleep and activity), while "M-C4" denotes integrated reasoning across all four domains, representing the most challenging setting.

We observe a clear trend: accuracy degrades markedly as the number of involved dimensions increases. Under context prompting, the average accuracy drops from 41.54% on single-dimension tasks to 30.74% on M-C2, and further to 23.42% on M-C4. Under database-augmented prompting, the decline is even steeper, from 30.84% down to 11.9% (M-C2) and 9.63% (M-C4). These results indicate that cross-dimensional reasoning particularly challenging, as models must not only capture fine-grained signals within each domain but also integrate interactions across lifestyle dimensions. This underscores the importance of advancing LLMs capability for cross-dimensional health reasoning.

For different user settings, from the performance of all LLMs, their performance on single-user tasks generally surpasses that on multi-user tasks (see Appendix H.3). In particular, with the *context-prompting setting*, multi-user reasoning tasks are infeasible because embedding large-scale multi-user data directly into the prompt exceeds the typical context-window limits. These experimental results suggest that, compared with individual-level reasoning, cross-user aggregation is more challenging, mainly due to the need for large-scale data integration and the management of complex inter-user relationships. This also highlights an important direction for future research.

## 5 CONCLUSION

We present MultiLifeQA, a large-scale cross-dimensional health QA dataset and benchmark with two evaluation settings and multiple metrics, and evaluate eight open-source and three proprietary LLMs. Experiments show that proprietary models outperform open-source ones, but they still exhibit clear limitations in cross-dimensional, long-horizon, and multi-user reasoning. Furthermore, understanding data relationships and generating complex reasoning remains a key bottleneck for current LLMs. Overall, the results highlight both potential and limitations of current LLMs, underscoring the value of MultiLifeQA as a health reasoning benchmark. The released code and dataset, together with an extensible framework and guidelines, support future research on new health data and tasks, pushing LLMs toward a more comprehensive paradigm of health reasoning.

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

## APPENDIX

## A    REPRODUCIBILITY STATEMENT

To ensure full reproducibility of our results, we release all code for data generation, prompt construction, and evaluation, together with processed datasets, schema definitions and guidelines at https://anonymous.4open.science/r/MultilifeQA-05D2. Our experimental settings, including model configurations, decoding parameters, and hardware environments, are described in detail in Appendix C. The exact prompt templates used in each evaluation setting are provided in Appendix E. We also specify the evaluation metrics and correctness criteria in the main text. With these resources, independent researchers can replicate our experiments and verify all reported numbers.

## B    THE USE OF LARGE LANGUAGE MODELS (LLMS)

We used LLMs to assist with grammar correction and polishing of the manuscript. All generated text was carefully checked and validated by the authors, who take full responsibility for the final content.

## C    IMPLEMENTATION DETAILS

We evaluate open-source models on two hardware setups: smaller models (deepseek-coder-1.3B, Llama-3.2-3B, Phi-3.5-mini-3.8B, Llama-3.1-8B, Mistral-v0.3-7B, Qwen2.5-7B, and gemma-2-IT-9B) were run on a single RTX 4090 GPU, while larger models (Qwen2.5-14B 4/8-bit, Qwen2.5-32B 4-bit, and Llama-3.1-70B 4-bit) were run on 4×A6000 GPUs; proprietary models (GPT-4o, Claude-3.7-Sonnet, Gemini 2.5 Pro) were accessed via official APIs. All experiments used a context length of 4096 tokens (or the maximum allowed if smaller). For *Context Prompting*, we set `max_new_tokens`=32. For *Database-augmented Prompting*, we used 480 tokens for the SQL generation step and 48 tokens for the answer generation step. All open-source models used deterministic decoding with temperature effectively disabled (greedy decoding) for reproducibility.

## D    QA DATASET GENERATION PIPELINE

### D.1    DETAILED DESCRIPTION OF THE DATASET GENERATION PIPELINE

To systematically evaluate large language models on multi-domain health reasoning tasks, we develop a general-purpose pipeline for automatic QA dataset construction. The pipeline integrates raw relational data from an open-source dataset AI4FoodDB into a unified framework and outputs question–answer (QA) pairs in three complementary forms: original QA, Context prompting and Database-augmented prompting. You can find our code and detailed guide at https://anonymous.4open.science/r/MultilifeQA-05D2. The generation pipeline proceeds as follow:

**Preparing Raw Data and MySQL Database.** Raw data sources, including structured CSV files and relational tables from AI4FoodDB and FoodNExtDB, are loaded into a MySQL database. The loading scripts define table schemas, map attributes across domains, and ensure consistent naming conventions. This step guarantees that both single-domain and cross-domain information can be queried with SQL.

**Template-based QA Generation.** Once the database is prepared, a set of extensible templates is applied to automatically generate questions and answers. Each template specifies the question structure, SQL retrieval logic, and answer derivation rules. The outputs are written into JSONL files, where each line corresponds to a QA pair.

**Processed Dataset Organization.** The generated QA pairs are organized into a hierarchical folder structure that reflects user scope (single-user vs. multi-user) and task complexity (single table vs. multi-table). In addition, we provide summary files (`all_prompts.jsonl`, `single_user.jsonl`, `multi_user.jsonl`) to facilitate direct evaluation. This structured design ensures that both context-based prompting and database-augmented prompting can be evaluated under consistent conditions.

Through this pipeline, we produce a large-scale, standardized dataset that covers diverse reasoning tasks across diet, activity, sleep, and emotion domains. The modularity of the pipeline also makes it suitable for adapting to new datasets or reasoning problems.

## D.2 GUIDELINE FOR EXTENSIBLE TEMPLATE FRAMEWORK

A key feature of our framework is its extensibility: researchers can expand more data and tasks by this pipeline. We summarize the guidelines for extending the template framework as follows.

**Prepare and Load Your Own Data.** Begin by organizing the target dataset into structured relational tables (e.g., CSV files). By modifying the provided loading scripts (`load_mysql_db.py`, `load_food_db.py`), users can map new attributes and table names into MySQL. Once loaded, the data becomes fully compatible with our pipeline.

**Define New Question Templates.** The framework is template-driven, which means that question styles and reasoning operations are explicitly defined. Users may: reuse the existing five categories (FQ, AS, CQ, NC, TA); extend to new categories such as causal reasoning, longitudinal trend analysis, or multi-hop inference; and modify the SQL generation logic or natural language phrasing to capture domain-specific constraints.

**Integrate with the Generation Scripts.** After defining new templates, they can be directly applied into the provided `build.py` script. After modifying key parts of this script, it will automatically generate questions, execute the corresponding *SQL* queries, and store the final QA pairs in `JSONL` format consistent with the existing dataset.

**Share and Evaluate.** By following the same structure, newly generated QA datasets can be seamlessly evaluated under both context prompting and database-augmented prompting. We encourage more researchers will share their extensions, which allows fair comparison across datasets while continually expanding the scope and difficulty of reasoning challenges.

**Summary.** The extensible template framework provides a principled yet flexible methodology for QA dataset construction. It not only ensures reproducibility and consistency across different prompting strategies, but also enables the community to explore new reasoning paradigms and domains while leveraging a common framework.

# E  PROMPT TEMPLATES

We design unified prompt templates for both *Context Prompting* and *Database-augmented Prompting* settings, as summarized in Tables 3 and 4. For Context Prompting, the prompt provides the question together with either a single compact TSV table or multiple compact tables for the same user, followed by explicit output requirements specifying the number and type of expected answers. For Database-augmented Prompting, the process is divided into two stages: *(i)* SQL generation, where the model is instructed to produce exactly one valid MySQL query based on the given schema, and *(ii)* reasoning after SQL execution, where the question, the executed SQL, and its result are provided, and the model is required to return the final answer in the prescribed format.

Table 3: Prompt Template for Context Prompting

---

**Prompt Template of Context Prompting**

---

SYSTEM:
You are a concise evaluator. Read the question and reply with ONLY the final answer (no explanation).
USER:

You are given **[TABLE_SCOPE]** TSV view(s) derived from **[TABLE_TYPE]**. Each view is restricted to a single entity (id). A `date` column is provided and should be treated as **[DATE_FIELD]**. **[SPECIAL_RULES]** Answer strictly with **[OUTPUT_FORMAT]**; do not include explanations.

Question: **[Question]**

=== BEGIN TABLE **[TABLE_NAME]** ===
**[TABLE1_TSV]**
=== END TABLE ===
... (If more than one table)
=== BEGIN TABLE **[TABLE_NAME]** ===
**[TABLEN_TSV]**
=== END TABLE ===

Output requirement: return **[Number_of_Answer]** value(s); types (ordered): {**[Answer_Type]**}; {**[Answer_Type]**}, ..., {**[Answer_Type]**}.

**Label definition:**

- **[Answer_Type]**: Choose from {"yes or no", "uid", "date", "datetime", "real number (two decimal)", "integer", "word"}.
- **[TABLE_SCOPE]**: "a compact TSV view" (single-table) or "compact TSV views from multiple relational tables" (multi-table).
- **[TABLE_TYPE]**: "a single relational table" or "multiple relational tables".
- **[DATE_FIELD]**: date normalization field, e.g. `DATE(ts)`, `DATE(start_time)`, `DATE(start_sleep)`, `DATE(record_ts)`, `DATE(night_end)`.
- **[SPECIAL_RULES]**: optional task-specific rules (e.g., deduplicate labels within a meal, count distinct meals containing a label, cross-reference by `id` and `date`).
- **[OUTPUT_FORMAT]**:
  - Single number (two decimals).
  - One word (yes/no; increase/decrease/same; categorical token).
  - Value+Date ("N on YYYY-MM-DD").
  - Semicolon-separated list of tokens.
  - Token–Number ("TOKEN; NUMBER").
  - Multiple Value+Date items (two/three; semicolon-separated).

---

Table 4: Prompt Templates of Database-augmented Prompting.

---

**Prompt Template of SQL generation**

---

SYSTEM:

You are an expert MySQL analyst. The database is already connected and available. Write ONE and only ONE read-only SQL query to answer the question. Constraints: MySQL dialect; SELECT/CTE only; no DDL/DML; no multiple statements; output SQL only.

USER:

Given the following MySQL table schema, write ONE SELECT statement to compute the data you need. Use DATE(ts) for date filtering if needed. Output SQL only.

Question: [Question]

Schema (DDL):

```sql
[Schema]
[Notes]
... (If more than one table)
[Schema]
[Notes]
```

[Note] The MySQL database is connected. Use DATE(ts) for day filtering. Output only one SQL statement; end with a semicolon; no backticks.

**Label definition:**

- [Schema]: SQL schema, e.g.: CREATE TABLE 'TABLE_NAME' (column_name column_type [constraints], ... , column_name column_type [constraints], PRIMARY KEY (column1, column2, ...))

- [Notes]: Special notes for this schema, e.g.:
    - Use DATE(column_name) for day filtering; group by column_name if needed.
    - Semantics: Deduplicate within a meal (id, ts, label). For "times" counts across a day/week, count distinct meals (ts) that include the label.

---

**Prompt Template of Reasoning after SQL execution**

---

SYSTEM:
You are a concise evaluator. You will see a question and the SQL result. Answer the question using ONLY the provided SQL result. Reply with ONLY the final answer (no explanations).

USER:

You will be given the SQL result for the question. Answer the question based on the SQL result, do not include explanations.

Question: [Question]

Output requirement: return [Number_of_Answer] value(s); types (ordered): {[Answer_Type]}; {[Answer_Type]}, ..., {[Answer_Type]}.

SQL used:
```sql
[Generated_SQL]
```
SQL result (rows=[Number_of_Rows]):
[SQL_Result]

**Label definition:**

- [Answer_Type]: Choose from {"yes or no", "uid", "date", "datetime", "real number (two decimal)", "integer", "word"}.
- Number_of_Rows: Number of rows fetched by SQL.
- [SQL_Result]: Results by executing the generated SQL.

---

# F  DATASET STATISTICS

To better understand the distribution of questions in MultiLifeQA , we summarize the dataset statistics under the two evaluation setups: *Context Prompting* and *Database-augmented Prompting*. Both setups are categorized along multiple dimensions, including question type, answer type, table setting, and domain coverage. The Database-augmented Prompting setup further distinguishes between single-user and multi-user cases.

Table 5 reports the detailed data distribution. With *Context Prompting*, the dataset contains a balanced set of reasoning categories that ensure coverage across different levels of difficulty. For question types, conditional queries (3,065; 22.8%) are the most frequent, followed by trend analysis (2,806; 20.9%), aggregation statistics (2,725; 20.3%), and numerical comparison (2,735; 20.3%), with fact-based queries slightly fewer (2,121; 15.8%). For answer types, single-number answers dominate (6,306; 46.9%), while yes/no (1,564; 11.6%), short text (1,308; 9.7%), pairwise (3,150; 23.4%), and multi-item answers (1,124; 8.4%) diversify the output formats. In terms of table settings, single-dimension tasks (4,029; 30.0%) coexist with a much larger proportion of cross-dimensional queries, most notably M-C2 (6,475; 48.1%), highlighting that the dataset places particular emphasis on cross-domain integration rather than isolated fact retrieval.

With *Database-augmented Prompting*, the distribution is larger but follows a similar trend. Conditional queries remain the largest category (5,134; 22.7%), with aggregation (4,732; 21.0%), numerical comparison (4,550; 20.2%), and trend analysis (4,755; 21.1%) also well represented. Answer types continue to be dominated by single numbers (10,642; 47.2%), with short text (4,061; 18.0%), pairwise (3,186; 14.1%), and multi-item answers (1,184; 5.2%) providing added diversity. For table settings, again the majority of queries involve multi-dimensional reasoning, including 6,475 M-C2 queries (28.7%) and 960 M-C4 queries (4.3%), emphasizing that the purpose of this benchmark is to evaluate the model's ability to reason across dimensions. Importantly, this setting also incorporates multi-user queries, such as 7,901 single-dimension (35.0%) and 1,220 M-C4 (5.4%) tasks, enabling evaluation of both individualized reasoning and population-level comparisons.

Overall, this design not only guarantees broad coverage of different reasoning difficulties but also emphasizes cross-domain and cross-user integration, ensuring that MultiLifeQA serves as a rigorous and comprehensive benchmark for evaluating LLMs in complex health reasoning.

Table 5: Data distribution of MultiLifeQA by setup, dimension, and category. Counts and percentages are computed within each category.

| Setup | Dimension | Category | Number | Percentage |
|---|---|---|---|---|
| Context Prompting | Question Type: 
 What kind of question in the dataset | FQ | 2121 | 15.8% |
| | | AS | 2725 | 20.3% |
| | | CQ | 3065 | 22.8% |
| | | NC | 2735 | 20.3% |
| | | TA | 2806 | 20.9% |
| | Answer Type: 
 What kind of answer in the dataset | Yes/No | 1564 | 11.6% |
| | | Single Number | 6306 | 46.9% |
| | | Short Text | 1308 | 9.7% |
| | | Pairwise Answer | 3150 | 23.4% |
| | | Multi-item Answer | 1124 | 8.4% |
| | Different Dimensions: 
 How many domains are covered | Single | 4029 | 30.0% |
| | | M-Sleep | 278 | 2.1% |
| | | M-Act | 1710 | 12.7% |
| | | M-C2 | 6475 | 48.1% |
| | | M-C4 | 960 | 7.1% |
| | Each Domain: 
 Counted if the domain appears in the question | Activity | 6472 | 48.1% |
| | | Sleep | 7155 | 53.2% |
| | | Emotion | 4920 | 36.6% |
| | | Diet | 4260 | 31.7% |
| Database-augmented Prompting | Question Type: 
 What kind of question in the dataset | FQ | 3402 | 15.1% |
| | | AS | 4732 | 21.0% |
| | | CQ | 5134 | 22.7% |
| | | NC | 4550 | 20.2% |
| | | TA | 4755 | 21.1% |
| | Answer Type: 
 What kind of answer in the dataset | Yes/No | 3500 | 15.5% |
| | | Single Number | 10642 | 47.2% |
| | | Short Text | 4061 | 18.0% |
| | | Pairwise Answer | 3186 | 14.1% |
| | | Multi-item Answer | 1184 | 5.2% |
| | Different Dimensions (single-user): 
 How many domains are covered | Single | 4029 | 17.8% |
| | | M-Sleep | 278 | 1.2% |
| | | M-Act | 1710 | 7.6% |
| | | M-C2 | 6475 | 28.7% |
| | | M-C4 | 960 | 4.3% |
| | Different Dimensions (multi-user): 
 How many domains are covered | Single | 7901 | 35.0% |
| | | M-C4 | 1220 | 5.4% |
| | Each Domain (single-user): 
 Counted if the domain appears in the question | Activity | 6472 | 28.7% |
| | | Sleep | 7155 | 31.7% |
| | | Emotion | 4920 | 21.8% |
| | | Diet | 4260 | 18.9% |
| | Each Domain (multi-user): 
 Counted if the domain appears in the question | Activity | 3531 | 15.6% |
| | | Sleep | 5267 | 23.3% |
| | | Emotion | 2030 | 9.0% |
| | | Diet | 1953 | 8.7% |

# G SOME EXAMPLES OF HEALTH QUERY REASONING

## G.1 EXAMPLES OF DIFFERENT DIMENSIONS FOR A SINGLE USER

Table 6: Examples of different dimensions for a single user in MultiLifeQA.

| Domain | Question Type | Sample |
|---|---|---|
| Diet | FQ | What subcategories of food did [uid] eat on [datetime]? |
| | AS | How many times did [uid] eat foods from category='Protein Sources' within one week, starting from [date]? |
| | CQ | How many days within a week did [uid] eat foods cooked in cooking_style='Oven-Baked', starting from [date]? |
| | NC | Which category of food did [uid] eat most frequently within one week, starting from [date]? |
| | TA | How many consecutive days did [uid] eat foods from category='Protein Sources', starting from [date]? |
| Physical Activity | FQ | On [datetime], how many steps did [uid] take during Walk? And on that day how many steps did A[uid] take in total? |
| | AS | What is the average distance covered and the average active_duration during Run of [uid] within one week, starting from [date]? |
| | CQ | How many days within one week did [uid] have resting_heart_rate lower than 61.02 or average_heart_rate during Run lower than 29, starting from [date]? |
| | NC | What was the highest distance and the highest active_duration during Workout within a week for [uid], and on which days did they occur, starting from [date]? |
| | TA | Did [uid]'s cardio_minutes, resting_heart_rate, and average_heart_rate during Run show the same trend (increase/decrease) on [date], compared to the previous day? |
| Sleep | FQ | On [datetime], what was [uid]'s rmssd during sleeping? And on that day what was his/her lower_bound_oxygen_saturation? |
| | AS | What is the total minutes_asleep and the average full_sleep_breathing_rate of [uid] within one week, starting from [date]? |
| | CQ | How many days within one week did [uid] have minutes_in_light_sleep fewer than 237.79 and light_sleep_breathing_rate lower than 16.1, starting from [date]? |
| | NC | Within one week starting from [date], which minimum was lower for [uid]: the sleep_average_oxygen_saturation or the full_sleep_breathing_rate? |
| | TA | How many consecutive days did [uid]'s minutes_asleep and full_sleep_breathing_rate both decrease, starting from [date]? |
| Emotion | FQ | What was the value of stress_score for [uid] on [date]? |
| | AS | What is the total exertion_points of [uid] within one week, starting from [date]? |
| | CQ | How many days within a week did [uid] have sleep_points greater than 17.14, starting from [date]? |
| | NC | How much higher was exertion_points for [uid] on [date] compared to the previous day? |
| | TA | How many consecutive days did [uid]'s stress_score decrease, starting from [date]? |
| Cross 2-domains | FQ | On [date], how many calories did [uid] burn and how long did he/she stay in bed? |
| | AS | What is the average calories burned and average minutes_in_bed of [uid] within one week, starting from [date]? |
| | CQ | How many days within one week did [uid] have nightly_temperature lower than 1.94 while also recording stress_score/sleep_points lower than 9.93, starting from [date]? |
| | NC | Within one week starting from [date], how many more very_active_minutes did [uid] record on the most active day compared to the least active day, and what was the most common food category on those days? |
| | TA | How many consecutive days did [uid]'s calories_minutes increase while his/her stress_score decreased, starting from [date]? |
| Cross 4-domains | FQ | On [date], what was [uid]'s very_active_minutes, what cooking_style did he/she consume most, what was his/her rmssd during sleep, and what were his/her responsiveness_points? |
| | AS | Within one week starting from [date], what was [uid]'s most frequent food category, his/her average minutes_in_rem sleep, and his/her average responsiveness_points? |
| | CQ | Within one week starting from [date], how many days did [uid] eat meals cooked with none more than 0 times, while getting rmssd greater than 36.43 and recording responsiveness_points greater than 21.53? |
| | NC | Within one week starting from [date], on the day when [uid] had the highest very_active_minutes, what was his/her most frequent food subcategory, and what were his/her minutes_in_rem sleep and stress_score? |
| | TA | How many consecutive days starting from [date] did [uid] increase his/her frequency of none meals and very_active_minutes, while minutes_asleep increased and stress_score decreased? |

## G.2 EXAMPLES OF DIFFERENT DIMENSIONS FOR MULTI-USER

Table 7: Examples of different dimensions for multi-user in MultiLifeQA.

| Domain | Question Type | Sample |
|---|---|---|
| Diet | FQ | What was the most common subcategory of food across all users on [date]? |
| | AS | Which user had the highest number of meals cooked in the same cooking_style within one week, starting from [date]? |
| | CQ | How many users consumed subcategory 'Juices' on [date]? |
| | NC | Which cooking_style was used most frequently across all users within one week, starting from [date]? |
| | TA | Was the most common category across all users on [date] the same as the previous day? |
| Physical Activity | FQ | Which user had the highest steps on [date]? |
| | AS | Which user had the highest lightly_active_minutes within one week, starting from [date]? |
| | CQ | How many users had sedentary_minutes greater than 372.31 on [date]? |
| | NC | Which activity type had the highest average steps across all users on [date]: running, walking, or cycling? |
| | TA | How many consecutive days was [uid]'s steps higher than the average across all users, starting from [date]? |
| Sleep | FQ | What was [uid]'s rank among all users for rmssd during sleeping on [date]? |
| | AS | What was the average nightly_temperature across all users within one week, starting from [date]? |
| | CQ | How many users had lower_bound_oxygen_saturation lower than 89.9 on [date]? |
| | NC | Within one week starting from [date], which minimum was lower for [uid]: the sleep_average_oxygen_saturation or the full_sleep_breathing_rate? |
| | TA | Was the average entropy across all users higher, lower, or the same within one week, starting from [date], compared to the previous week? |
| Emotion | FQ | Which user had the highest stress_score on [date]? |
| | AS | What was the median exertion_points across all users within one week, starting from [date]? |
| | CQ | How many users had stress_score lower than 390.4 within one week, starting from [date]? |
| | NC | Was [uid]'s stress_score lower than the median across all users on [date]? |
| | TA | How many consecutive days did the average sleep_points across all users increase, starting from [date]? |
| Cross 4-domains | FQ | Which user consumed protein sources category and also had the highest resting_heart_rate on [date]? |
| | AS | Which user had the most days consuming meat and also the highest average steps within one week, starting from [date]? |
| | CQ | How many users had at least 5 days with steps $\geq$ population daily P70 within one week, starting from [date]? |
| | NC | Within one week starting [date], was the average resting_heart_rate lower among high-oxygen users (daily oxygen $\geq$ P70) than among low-oxygen users ($\leq$ P30)? |
| | TA | How many consecutive days did the average sleep_points across all users increase, starting from [date]? |

# H   SUPPLEMENTARY EXPERIMENTAL RESULTS

## H.1   RESULTS BY QUESTION TYPE AND ANSWER TYPE.

We further analyze the performance of all LLMs by splitting results across different question types and answer types, as summarized in Table 8. The results reveal clear differences across categories. Aggregate statistics (AS) and numeric comparison (NC) questions are generally more challenging, with low accuracy across open-source models, while GPT-4o achieves significantly higher performance. In contrast, fact query (FQ), conditional query (CQ) and trend analysis (TA) show higher variance across models, reflecting the difficulty of reasoning over multiple conditions.

On the answer side, binary (Yes/No) and single-number questions are relatively easier for most models, while pairwise and multi-item answers exhibit very low accuracy, highlighting that generating multiple related outputs remains a key challenge. When the answer type is short and simple, most LLMs can provide the correct response as long as the SQL query retrieves the right information, as reflected in the high Acc/EX scores. However, for pairwise and multi-item answers, Acc/EX remains low, indicating that increasing complexity not only makes SQL generation more difficult but also makes it harder for most LLMs to identify the correct answer from what retured by SQL when relying solely on the question and the SQL query itself. These findings underscore the limitations of current LLMs in handling complex multi-condition reasoning and compositional answer generation.

Table 8: Unified results by Question Type and Answer Type. Left: model/setup/metric; Right: statistics across multiple categories. Question Type (FQ/AS/CQ/NC/TA) and Answer Type (Yes/No, Single Number, Short Text, Pairwise Answer, Multi-item Answer). For Database-augmented Prompting, we show both Acc and Acc/EX as separate rows.

| Model | Setup | Metric | Question Type | | | | | Answer Type | | | | |
|---|---|---|---|---|---|---|---|---|---|---|---|---|
| | | | FQ | AS | CQ | NC | TA | Yes/No | Single Number | Short Text | Pairwise Answer | Multi-item Answer |
| **Open Source LLMs** | | | | | | | | | | | | |
| deepseek-coder-1.3B | Context Prompting | Acc | 0.47 | 0.04 | 0.0 | 0.37 | 4.78 | 8.57 | 0.32 | 0.76 | 0.0 | 0.0 |
| | Database-augmented Prompting | Acc | 0.06 | 0.08 | 2.90 | 1.60 | 1.18 | 3.00 | 1.68 | 0.0 | 0.0 | 0.0 |
| | | Acc/EX | 0.21 | 0.20 | 2.15 | 10.55 | 11.95 | 14.54 | 1.76 | 0.0 | 0.0 | 0.0 |
| Llama-3.2-3B | Context Prompting | Acc | 10.27 | 1.17 | 38.95 | 10.75 | 34.82 | 33.06 | 29.92 | 21.33 | 0.98 | 0.09 |
| | Database-augmented Prompting | Acc | 17.72 | 9.82 | 19.81 | 11.63 | 9.00 | 17.78 | 16.04 | 17.43 | 0.13 | 0.0 |
| | | Acc/EX | 68.69 | 84.64 | 79.19 | 52.87 | 52.32 | 52.05 | 81.19 | 69.60 | 6.67 | 0.0 |
| Phi-3.5-mini-3.8B | Context Prompting | Acc | 13.96 | 2.53 | 28.42 | 19.96 | 35.10 | 41.62 | 27.70 | 23.70 | 1.84 | 0.09 |
| | Database-augmented Prompting | Acc | 25.10 | 10.65 | 24.68 | 13.01 | 9.06 | 9.37 | 21.43 | 25.46 | 0.16 | 0.0 |
| | | Acc/EX | 69.09 | 91.94 | 93.95 | 65.98 | 61.87 | 51.17 | 90.33 | 86.94 | 1.73 | 0.0 |
| Mistral-v0.3-7B | Context Prompting | Acc | 24.47 | 2.83 | 49.43 | 13.89 | 59.69 | 48.66 | 44.61 | 28.13 | 6.79 | 0.89 |
| | Database-augmented Prompting | Acc | 20.99 | 6.78 | 11.24 | 9.01 | 0.34 | 8.63 | 10.75 | 14.55 | 0.03 | 0.0 |
| | | Acc/EX | 82.45 | 77.48 | 81.69 | 59.76 | 55.00 | 55.21 | 83.92 | 88.50 | 1.96 | 0.0 |
| Qwen2.5-7B | Context Prompting | Acc | 19.38 | 5.21 | 70.34 | 24.42 | 73.56 | 63.38 | 60.69 | 42.66 | 1.78 | 0.36 |
| | Database-augmented Prompting | Acc | 29.78 | 17.37 | 36.21 | 12.90 | 11.80 | 11.71 | 27.07 | 32.55 | 6.06 | 3.04 |
| | | Acc/EX | 83.57 | 94.19 | 95.15 | 65.86 | 70.03 | 60.21 | 97.53 | 80.05 | 63.46 | 29.51 |
| Llama-3.1-8B | Context Prompting | Acc | 20.93 | 2.83 | 24.67 | 17.92 | 36.03 | 56.78 | 21.31 | 25.31 | 5.24 | 4.45 |
| | Database-augmented Prompting | Acc | 24.96 | 11.39 | 30.46 | 28.13 | 13.21 | 24.68 | 25.93 | 28.27 | 1.51 | 3.46 |
| | | Acc/EX | 67.44 | 82.74 | 97.77 | 71.32 | 66.27 | 58.14 | 97.35 | 82.89 | 14.33 | 18.46 |
| gemma-2-IT-9B | Context Prompting | Acc | 23.90 | 4.29 | 33.18 | 14.15 | 44.87 | 22.44 | 35.70 | 37.92 | 6.00 | 0.0 |
| | Database-augmented Prompting | Acc | 19.08 | 9.62 | 17.04 | 17.43 | 10.71 | 15.97 | 15.64 | 24.43 | 1.69 | 1.01 |
| | | Acc/EX | 59.43 | 51.15 | 55.78 | 47.13 | 41.25 | 35.00 | 58.05 | 66.60 | 18.93 | 7.69 |
| Qwen2.5-14B (4-bit) | Context Prompting | Acc | 17.49 | 6.24 | 62.97 | 30.24 | 71.45 | 56.39 | 59.56 | 49.84 | 0.35 | 0.18 |
| | Database-augmented Prompting | Acc | 22.87 | 13.10 | 22.38 | 15.78 | 9.13 | 11.60 | 16.89 | 30.16 | 6.94 | 4.22 |
| | | Acc/EX | 61.01 | 60.59 | 50.10 | 49.20 | 63.16 | 49.21 | 50.40 | 68.91 | 58.75 | 27.22 |
| Qwen2.5-14B (8-bit) | Context Prompting | Acc | 60.54 | 8.29 | 75.89 | 35.65 | 75.52 | 62.72 | 66.97 | 52.14 | 26.19 | 19.48 |
| | Database-augmented Prompting | Acc | 29.59 | 17.05 | 29.59 | 18.31 | 5.78 | 9.26 | 22.35 | 33.02 | 7.31 | 2.87 |
| | | Acc/EX | 84.23 | 78.30 | 84.23 | 74.68 | 72.30 | 62.19 | 82.44 | 83.56 | 63.66 | 21.85 |
| Qwen2.5-32B (4-bit) | Context Prompting | Acc | 65.48 | 9.80 | 69.98 | 41.94 | 81.90 | 71.99 | 67.46 | 56.12 | 28.38 | 21.17 |
| | Database-augmented Prompting | Acc | 29.75 | 19.65 | 46.67 | 25.49 | 12.34 | 17.71 | 34.03 | 37.85 | 8.29 | 3.63 |
| | | Acc/EX | 64.84 | 79.84 | 79.71 | 73.18 | 76.50 | 69.70 | 79.19 | 82.54 | 48.95 | 20.28 |
| Llama-3.1-70B | Context Prompting | Acc | 47.52 | 8.00 | 40.10 | 34.59 | 73.02 | 65.15 | 48.16 | 49.92 | 19.81 | 10.41 |
| | Database-augmented Prompting | Acc | 15.26 | 7.78 | 22.22 | 16.55 | 7.57 | 11.80 | 15.28 | 26.08 | 1.35 | 0.0 |
| | | Acc/EX | 52.43 | 56.29 | 67.73 | 54.93 | 51.04 | 46.30 | 63.79 | 74.90 | 12.91 | 0.0 |
| **Proprietary LLMs** | | | | | | | | | | | | |
| Gemini 2.5 Lite | Context Prompting | Acc | 59.74 | 9.80 | 54.78 | 33.42 | 67.74 | 48.85 | 56.56 | 52.68 | 25.71 | 17.62 |
| | Database-augmented Prompting | Acc | 39.04 | 34.86 | 54.32 | 44.46 | 36.23 | 44.89 | 45.49 | 50.26 | 9.26 | 5.41 |
| | | Acc/EX | 82.92 | 73.75 | 98.30 | 76.89 | 78.88 | 70.23 | 97.92 | 84.75 | 40.31 | 22.86 |
| Claude-3-haiku | Context Prompting | Acc | 45.73 | 7.01 | 52.20 | 23.95 | 47.51 | 41.05 | 46.29 | 39.98 | 17.27 | 10.77 |
| | Database-augmented Prompting | Acc | 28.22 | 15.49 | 45.62 | 29.01 | 26.46 | 34.31 | 32.78 | 44.74 | 1.76 | 4.31 |
| | | Acc/EX | 64.95 | 79.12 | 98.35 | 64.25 | 63.88 | 56.73 | 94.40 | 90.28 | 9.56 | 21.25 |
| GPT-4o | Context Prompting | Acc | 69.78 | 15.63 | 71.45 | 49.58 | 79.08 | 75.06 | 68.31 | 61.62 | 33.87 | 28.11 |
| | Database-augmented Prompting | Acc | 42.50 | 26.45 | 58.47 | 31.31 | 14.89 | 19.29 | 41.50 | 50.41 | 15.63 | 16.98 |
| | | Acc/EX | 93.74 | 98.89 | 99.26 | 87.52 | 96.85 | 77.59 | 99.23 | 99.17 | 86.04 | 87.78 |

## H.2 THE IMPACT OF MODEL SIZE AND QUANTIZATION

Table 9 summarizes the performance of Qwen2.5 variants across different model scales and quantization settings. The results show that model size and precision both have a clear influence on accuracy. Larger models such as Qwen2.5-32B (4-bit) and Qwen2.5-14B (8-bit) achieve higher performance under context prompting compared to the smaller 7B variant, indicating that larger models size improve reasoning and answer generation across most question and answer types.

However, quantization introduces trade-offs. The 14B 4-bit model shows a significant drop in accuracy relative to its 8-bit counterpart, even being surpassed by the 7B model. These patterns highlight that both scaling up model size and preserving sufficient numerical precision are critical for reliable performance in multi-table reasoning tasks.

Table 9: Results of Qwen2.5 variants across scales and quantization.

| Model | Setup | Metric | Question Type | | | | | Answer Type | | | | | Overall |
|---|---|---|---|---|---|---|---|---|---|---|---|---|---|
| | | | FQ | AS | CQ | NC | TA | Yes/No | Single Number | Short Text | Pairwise Answer | Multi-item Answer | |
| Qwen2.5-7B | Context Prompting | Acc | 19.38 | 5.21 | 70.34 | 24.42 | 73.56 | 63.38 | 60.69 | 42.66 | 1.78 | 0.36 | 40.45 |
| | Database-augmented Prompting | Acc | 29.78 | 17.37 | 36.21 | 12.90 | 11.80 | 11.71 | 27.07 | 32.55 | 6.06 | 3.04 | 21.45 |
| | | Acc/EX | 83.57 | 94.19 | 95.15 | 65.86 | 70.03 | 60.21 | 97.53 | 80.05 | 63.46 | 29.51 | 84.78 |
| Qwen2.5-14B (4-bit) | Context Prompting | Acc | 17.49 | 6.24 | 62.97 | 30.24 | 71.45 | 56.39 | 59.56 | 49.84 | 0.35 | 0.18 | 38.42 |
| | Database-augmented Prompting | Acc | 22.87 | 13.10 | 22.38 | 15.78 | 9.13 | 11.60 | 16.89 | 30.16 | 6.94 | 4.22 | 16.39 |
| | | Acc/EX | 61.01 | 60.59 | 50.10 | 49.20 | 63.16 | 49.21 | 50.40 | 68.91 | 58.75 | 27.22 | 54.82 |
| Qwen2.5-14B (8-bit) | Context Prompting | Acc | 60.54 | 8.29 | 75.89 | 35.65 | 75.52 | 62.72 | 66.97 | 52.14 | 26.19 | 19.48 | 51.51 |
| | Database-augmented Prompting | Acc | 29.59 | 17.05 | 29.59 | 18.31 | 5.78 | 9.26 | 22.35 | 33.02 | 7.31 | 2.87 | 19.10 |
| | | Acc/EX | 84.23 | 78.30 | 84.23 | 74.68 | 72.30 | 62.19 | 82.44 | 83.56 | 63.66 | 21.85 | 77.82 |
| Qwen2.5-32B (4-bit) | Context Prompting | Acc | 65.48 | 9.80 | 69.98 | 41.94 | 81.90 | 71.99 | 67.46 | 56.12 | 28.38 | 21.17 | 53.86 |
| | Database-augmented Prompting | Acc | 29.75 | 19.65 | 46.67 | 25.49 | 12.34 | 17.71 | 34.03 | 37.85 | 8.29 | 3.63 | 26.95 |
| | | Acc/EX | 64.84 | 79.84 | 79.71 | 73.18 | 76.50 | 69.70 | 79.19 | 82.54 | 48.95 | 20.28 | 75.24 |

## H.3 RESULTS WITH VARYING USER SETTINGS AND DIMENSIONS.

Table 10 reports results across different user settings and levels of table complexity. The Single setup corresponds to a single table, M-Sleep aggregates five sleep-related tables, M-Activity combines three activity-related tables, and M-C4 involves ten tables spanning sleep, activity, diet, and emotion domains.

The results show a clear trend: accuracy generally decreases as the number of tables and domain coverage increase. Most models achieve their highest performance under the Single-table setting, while performance drops notably in M-Sleep and M-Activity, and further degrades in the more complex M-C4 setting. This reflects the increasing difficulty of reasoning over larger and more heterogeneous schemas.

Database-augmented prompting improves execution validity (Acc/EX), but raw accuracy often lags behind context prompting, particularly in multi-table scenarios. Proprietary models such as GPT-4o and Gemini 2.5 Lite maintain stronger robustness under complex settings, whereas open-source models experience sharper performance degradation. Overall, these findings highlight that the combination of multi-table integration and multi-domain reasoning substantially increases task difficulty, underscoring the need for future methods that can better handle schema complexity and cross-domain reasoning.

Table 10: Results under different user settings and dimensions. (Single = Single dimension; M-Sleep = multi-dimension within sleep domain; M-Act = multi-dimension within activity domain; M-C2 = multi-dimension across two domains; M-C4 = multi-dimension across four domains.)

| Model | Setup | Metric | Single-user | | | | | Multi-user | |
|---|---|---|---|---|---|---|---|---|---|
| | | | Single | M-Sleep | M-Activity | M-C2 | M-C4 | Single | M-C4 |
| **Open Source LLMs** | | | | | | | | | |
| deepseek-coder-1.3B | Context Prompting | Acc | 0.77 | 0.72 | 3.57 | 0.37 | 2.92 | / | / |
| | Database-augmented Prompting | Acc | 2.90 | 2.88 | 0.64 | 1.05 | 0.00 | 1.01 | 0.00 |
| | | Acc/EX | 2.39 | 21.05 | 1.59 | 5.95 | 0.00 | 4.17 | 0.00 |
| Llama-3.2-3B | Context Prompting | Acc | 22.53 | 10.07 | 23.68 | 16.90 | 29.17 | / | / |
| | Database-augmented Prompting | Acc | 17.42 | 0.00 | 4.21 | 6.17 | 0.63 | 21.58 | 12.87 |
| | | Acc/EX | 66.78 | 0.00 | 58.62 | 58.95 | 75.00 | 71.54 | 68.11 |
| Phi-3.5-mini-3.8B | Context Prompting | Acc | 28.10 | 14.75 | 26.67 | 17.27 | 2.08 | / | / |
| | Database-augmented Prompting | Acc | 32.06 | 0.36 | 3.33 | 4.99 | 2.71 | 23.72 | 6.15 |
| | | Acc/EX | 86.45 | 1.35 | 53.40 | 50.09 | 92.86 | 83.37 | 56.15 |
| Mistral-v0.3-7B | Context Prompting | Acc | 34.60 | 24.82 | 42.46 | 29.30 | 8.33 | / | / |
| | Database-augmented Prompting | Acc | 19.71 | 0.00 | 0.70 | 0.45 | 0.31 | 15.15 | 0.25 |
| | | Acc/EX | 77.57 | 0.00 | 100.00 | 18.84 | 100.00 | 76.34 | 75.00 |
| Qwen2.5-7B | Context Prompting | Acc | 45.00 | 18.70 | 48.54 | 40.23 | 14.69 | / | / |
| | Database-augmented Prompting | Acc | 38.37 | 21.22 | 15.43 | 12.74 | 2.81 | 24.52 | 15.00 |
| | | Acc/EX | 86.28 | 83.10 | 88.00 | 86.33 | 67.50 | 83.97 | 74.59 |
| Llama-3.1-8B | Context Prompting | Acc | 28.07 | 19.06 | 21.11 | 16.73 | 15.63 | / | / |
| | Database-augmented Prompting | Acc | 35.15 | 4.68 | 8.95 | 12.91 | 4.48 | 27.12 | 20.98 |
| | | Acc/EX | 91.01 | 13.27 | 60.96 | 68.29 | 48.31 | 80.43 | 69.32 |
| gemma-2-IT-9B | Context Prompting | Acc | 42.59 | 11.15 | 38.25 | 13.68 | 0.00 | / | / |
| | Database-augmented Prompting | Acc | 22.01 | 4.32 | 6.08 | 7.27 | 3.02 | 21.26 | 9.03 |
| | | Acc/EX | 53.27 | 28.57 | 32.10 | 34.46 | 26.61 | 61.74 | 47.74 |
| Qwen2.5-14B (4-bit) | Context Prompting | Acc | 52.54 | 14.75 | 45.44 | 34.89 | 11.35 | / | / |
| | Database-augmented Prompting | Acc | 22.46 | 5.03 | 10.79 | 10.19 | 18.65 | 20.01 | 21.15 |
| | | Acc/EX | 45.26 | 54.72 | 23.99 | 55.84 | 89.95 | 58.23 | 93.39 |
| Qwen2.5-14B (8-bit) | Context Prompting | Acc | 53.26 | 44.60 | 53.98 | 51.43 | 42.40 | / | / |
| | Database-augmented Prompting | Acc | 36.01 | 4.68 | 3.16 | 9.22 | 10.63 | 24.00 | 16.23 |
| | | Acc/EX | 85.04 | 31.71 | 26.70 | 82.70 | 26.70 | 75.09 | 94.33 |
| Qwen2.5-32B (4-bit) | Context Prompting | Acc | 59.09 | 48.92 | 54.80 | 51.14 | 50.10 | / | / |
| | Database-augmented Prompting | Acc | 39.31 | 2.52 | 26.90 | 16.01 | 20.31 | 30.82 | 30.08 |
| | | Acc/EX | 75.12 | 5.79 | 85.29 | 65.49 | 87.44 | 78.70 | 88.97 |
| Llama-3.1-70B (4-bit) | Context Prompting | Acc | 51.28 | 35.25 | 42.51 | 33.76 | 38.85 | / | / |
| | Database-augmented Prompting | Acc | 16.95 | 0.72 | 4.39 | 7.29 | 12.71 | 20.39 | 14.43 |
| | | Acc/EX | 54.18 | 11.76 | 24.15 | 46.18 | 62.56 | 69.19 | 73.13 |
| **Proprietary LLMs** | | | | | | | | | |
| Gemini 2.5 Lite | Context Prompting | Acc | 55.35 | 46.76 | 44.21 | 40.20 | 32.19 | / | / |
| | Database-augmented Prompting | Acc | 54.13 | 18.71 | 25.38 | 32.09 | 22.81 | 42.40 | 40.81 |
| | | Acc/EX | 93.47 | 41.60 | 80.00 | 77.86 | 60.07 | 84.10 | 78.25 |
| Claude-3-haiku | Context Prompting | Acc | 45.79 | 42.81 | 42.81 | 30.01 | 22.71 | / | / |
| | Database-augmented Prompting | Acc | 39.54 | 11.15 | 23.74 | 23.14 | 12.19 | 33.73 | 19.51 |
| | | Acc/EX | 80.68 | 28.85 | 69.28 | 62.91 | 55.39 | 89.02 | 70.21 |
| GPT-4o | Context Prompting | Acc | 62.64 | 47.12 | 54.67 | 54.51 | 57.39 | / | / |
| | Database-augmented Prompting | Acc | 55.67 | 32.73 | 31.40 | 23.04 | 23.54 | 36.27 | 31.31 |
| | | Acc/EX | 98.40 | 89.22 | 99.63 | 93.40 | 99.12 | 94.71 | 90.63 |

