# OpenReview forum: "MultiLifeQA: A Multidimensional Lifestyle Question Answering Benchmark for Comprehensive Health Reasoning with LLMs"
_ICLR.cc/2026/Conference — ICLR 2026 Conference Withdrawn Submission_

### Official Review · Reviewer_LveZ · 2025-10-31

**Soundness:** 3
**Presentation:** 2
**Contribution:** 2
**Rating:** 2
**Confidence:** 3

**Summary:**

The paper introduces MultiLifeQA, a large-scale benchmark designed to evaluate LLMs’ reasoning abilities over multidimensional lifestyle health data. Built on the AI4FoodDB dataset, it covers four lifestyle domains—diet, activity, sleep, and emotion—spanning 22,573 QA pairs across five reasoning types (Fact Query, Aggregated Statistics, Numeric Comparison, Conditional Query, Trend Analysis) and both single-user and multi-user settings. The authors provide two evaluation frameworks: (1) Context Prompting, where health data are directly embedded in prompts, and (2) Database-Augmented Prompting, where the LLM must generate and execute SQL queries before reasoning. They test 11 LLMs (8 open-source, 3 proprietary) and propose fine-grained metrics—Accuracy, SQL Validity, Execution Accuracy, and Acc/EX—to dissect model performance. Results reveal that proprietary models (GPT-4o, Gemini) outperform open-source ones but all models struggle with cross-dimensional, multi-user, and long-horizon reasoning, particularly in generating executable SQL and aggregating over large temporal windows

**Strengths:**

1. Novel and comprehensive benchmark: This is the first dataset unifying diet, activity, sleep, and emotion reasoning under both single- and multi-user scenarios, offering a rigorous framework for assessing integrated health reasoning. 2. Methodologically solid evaluation design: The dual setup (context vs. database prompting) and fine-grained metrics (VA, EX, Acc/EX) enable granular diagnosis of reasoning weaknesses in LLMs. 3. Strong empirical foundation and reproducibility: The benchmark is grounded on a validated health dataset (AI4FoodDB) and released with a reproducible pipeline, enhancing future extensibility and transparency.

**Weaknesses:**

1. My major concern is that the five listed tasks - Fact Query (FQ), Aggregated Statistics (AS), Numeric comparison (NC), Conditional Query (CQ) and Trend Analysis (TA) - are generally considered fact questions and not really related to the usage of domain knowledge. Therefore, it seems the benchmark still challenges LLMs mostly on their querying and context understanding capabilities, and less about domain understanding. In this case, the point of developing a domain specific benchmark is not really clear - Do LLMs fail dramatically on this benchmark because it's trained on general corpus?
2. Another concern is that the ground truth generation is largely based on "a programmatic framework", meaning the ground truth is generated mechanically. Though it's followed by a so-called human check by the authors, what's point of using LLM if this task can already be largely solved by a programmatic framework? Or would the authors prove that after fine-tuning on this benchmark, LLM's domain capabilities will boost significantly?
3. Overall, I don't see the unique challenges in this domain task, and I don't see the novelty of creating such a benchmark (though of course this is the first one in this  narrow domain as the authors claimed). As a comparing example, I would consider a domain benchmark quite meaningful, if it's manually curated by domain experts on a task only human experts can perform well, and we benchmark LLM's capabilities and hope one day some models can handle the tasks well.

**Questions:**

N/A

---

### Official Review · Reviewer_4Cu2 · 2025-10-31

**Soundness:** 2
**Presentation:** 1
**Contribution:** 1
**Rating:** 2
**Confidence:** 3

**Summary:**

The authors provide a benchmark for data-backed QA on personal health data. They take a standard dataset of personal health information and augment it by generating templated QA pairs. The explore several different forms of questions and benchmark LMs both on their ability to use databases and answer questions with information in their contexts.

**Strengths:**

- I have not seen multi-user questions from similar papers and so this is a novel contribution.
- Other related works like "Transforming wearable data into health insights using large language model agents" (Merrill et al. 2024) do not publish data

**Weaknesses:**

- It's a major limitation that the questions and ground-truth answers are themselves LM-generated. How do we know that LM-generated questions are realistic (i.e. that a human would ask these questions in a real deployment)? Furthermore, if an LM generated the ground truth in the first place then is it particularly illuminating that and LM is able to answer them during evaluation? I don't think so. I also have concerns about the evaluation setting, see below.
- The paper could be improved by justifying why we need a health-specific QA benchmark. Are there particular properties of personal health questions that mean that they need their own benchmark? If so, does this benchmark provide them?
- The paper is not particularly well-written. There are many run-on sentences and the language is altogether verbose.

**Questions:**

- How exactly was the quality of the ground truth answers verified? Were they manually inspected, or was the code checked for accuracy? Or something else?
- Might there be multiple plausible answer to a give query? For example, if I ask "how much did I sleep last week" this query could correspond to the last seven days, or the last calendar week Saturday-Saturday, or Sunday-Sunday. I wonder if this explains why performance on aggregation statistics is so poor.
- I'm very surprised that these frontier models have such low execution accuracy. How did the models fail? Did they e.g. hallucinate column names or datatypes?
- Do stronger models like Claude 4.1 Opus come close to saturating this benchmark? I think that without testing these you cannot claim that "proprietary models ... still exhibit clear limitations in cross-dimensional, long-horizon, and multi-user reasoning"

---

### Official Review · Reviewer_p4Lp · 2025-10-31

**Soundness:** 2
**Presentation:** 1
**Contribution:** 2
**Rating:** 2
**Confidence:** 3

**Summary:**

This paper introduces MultiLifeQA, a large-scale benchmark designed to assess large language models (LLMs) on health oriented data tables. Built upon the AI4FoodDB dataset, MultiLifeQA spans diet, activity, sleep, and emotion and includes 22,573 QA pairs across both single-user and multi-user settings. The questions are constructed via templates.

The authors find that when LMs such as GPT-4o and Gemini 2.5 Flash-Lite are especially poor on performing aggregation on entire data tables.

**Strengths:**

S1. MultiLifeQA integrates multiple lifestyle domains and creates a large dataset of health question answering over tabular/database data.

S2. The baselines included are reasonable and a good range of LLMs are included.

**Weaknesses:**

W1. My main concern with the work is that it is pitched as a benchmark for multidimensional lifestyle health reasoning but the at the core when looking at the questions (e.g., Table 6 and Table 7) the benchmark essentially contains questions over data tables. Questions such as "what was the value of stress score for [uid] on [date]?" to me is less so about health reasoning but basic reasoning over data tables. Is this fundamentally different from a question that was "What was the value of cars manufactured for company [X] on [date]?" on a similar table of car manufacturing. It is unclear what about the health data at least framed in this work makes it more challenging or about health reasoning.

Therefore findings such as "aggregation statistics are the most challenging, with average accuracies of only 5.98% (CP) and 14.2% (DP)" seem more about a LLMs capability to do table aggregation vs. actual health reasoning. If this is the case, there are plenty of works that evaluate this already [1-5]

W2. The paper currently lacks critical detail that help readers ensure the quality of the dataset.  First with respect to human validation, line 234 mentions that all questions were inspected. Does this mean all 22,574 questions were manually reviewed? What is the human verification process? Who were the humans involved?
Second, what are the statistics of the database being passed in during experiments? What are the table sizes / number of tables for the queries.

W3. This is more so for clarity of presentation. The baselines (i.e., context prompting and data-base augmented prompting) are not on the same split of the data. Thus, when reporting the results in line 114, Table 2, Figure 4, Figure 5 etc. it can be confusing/misleading as the raw numbers are not comparable between the two baselines.

W4. Minor, none of the frontier models (i.e., GPT-5, larger reasoning/thinking models) are evaluated so it is unclear whether the findings are actually consistent with modern LLM capabilities. My main concern with this is whether the dataset would be informative if used today to evaluate current frontier LLMs.

[1] How well do LLMs reason over tabular data, really? Cornelius Wolff and Madelon Hulsebos, arXiv preprint arXiv:2505.07453 (2025).

[2] Tabular Representation, Noisy Operators, and Impacts on Table Structure Understanding Tasks in LLMs, Ananya Singha et al., Table Representation Learning workshop at NeurIPS 2023.

[3] RADAR: Benchmarking Language Models on Imperfect Tabular Data, Ken Gu et al., NeurIPS D&B Track 2025

[4] TableBench: A Comprehensive and Complex Benchmark for Table Question Answering. Xianjie Wu et al., AAAI 2025

[5] FREB-TQA: A Fine-Grained Robustness Evaluation Benchmark for Table Question Answering. Zhou et al., NAACL 2024

**Questions:**

See above.

---

### Official Review · Reviewer_cYFX · 2025-11-01

**Soundness:** 1
**Presentation:** 2
**Contribution:** 1
**Rating:** 2
**Confidence:** 4

**Summary:**

The paper introduces a dataset for lifestyle-related question answering using template generation. It offers both single-user and multi-user scenarios. The paper also introduces hand-crafted dimensions (diet, sleep, activity, and emotion) and task categories (fact query, aggregated statistics, numeric comparison, conditional query, and trend analysis). It aims to support long-term, multidimensional health reasoning.

**Strengths:**

- 22,573 questions/answers, four dimensions (diet, activity, sleep, and emotion), five task categories (fact query, aggregated statistics, numeric comparison, conditional query, and trend analysis)
- There is a need for a lifestyle-related question answering benchmark
- Tested on 9 open-source LLMs and 3 proprietary LLMs
- Good visualisations and some interesting insights

**Weaknesses:**

- The dataset is based on template generations, raising many unanswered questions. The examples of health query reasoning in Appendix G are well documented but expose serious issues with synthetic data. In real-life contexts, it is important to understand when, why, and how someone asked the questions. In what context or how often, someone would ask "What subcategories of food did [uid] eat on [datetime]?" or "What was the most common subcategory of food across all users on [date]?" ?
- No/limited information on human verification/validations
- Hand-crafted dimensions remain limited - work/transportation/recreation/social connections are also important
- Linked to code/SQL generation tasks - but limited experiments on sophisticated LLM techniques, such as reasoning, tool-use, ...
- Limited novelty and can be easily replicated

**Questions:**

- How do you quantify the quality and diversity of the dataset? Why did the authors generate 22K questions, but not 10K, 100K, or 1M? Are these questions/answers realistic? How can we mitigate the potential negative impacts of the MultiLifeQA dataset on high-quality, data-driven, manually created datasets?
- How was the data validated? What are the human annotation processes? Agreement levels?
- How about structural, temporal, and personal dimensions of lifestyle data?
- Would it be possible to also investigate multi-turn QA, advanced reasoning, and tool-use?

---

### Note · Authors · 2025-12-03

I have read and agree with the venue's withdrawal policy on behalf of myself and my co-authors.